# Cognitive Impairment in Anti-Phospholipid Syndrome and Anti-Phospholipid Antibody Carriers

**DOI:** 10.3390/brainsci12020222

**Published:** 2022-02-05

**Authors:** Fadi Hassan, Mohammad E. Naffaa, Amir Saab, Chaim Putterman

**Affiliations:** 1Rheumatology Unit, Galilee Medical Center, Naharyia 2210001, Israel; fadihh@gmail.com; 2Azrieli Faculty of Medicine, Bar-Ilan University, Safed 1311502, Israel; amirs@gmc.gov.il (A.S.); chaim.putterman@einsteinmed.edu (C.P.); 3Internal Medicine “E”, Galilee Medical Center, Naharyia 2210001, Israel; 4Division of Rheumatology, Albert Einstein College of Medicine, Bronx, NY 10461, USA; 5Research Institute, Galilee Medical Center, Naharyia 2210001, Israel

**Keywords:** cognitive impairment, dementia, anti-phospholipid syndrome, anti-phospholipid carrier

## Abstract

Cognitive impairment is frequently reported among anti-phospholipid syndrome (APS) patients as well as anti-phospholipid antibody (aPL) carriers, but it is less studied than other manifestations of this condition. Moreover, the exact prevalence of cognitive impairment in these patients has not been accurately determined, mainly due to inconsistency in the tools used to identify impairment, small sample sizes, and variability in the anti-phospholipid antibodies measured and positivity cutoffs. The notion of a direct pathogenic effect is supported by the observation that the higher the number of aPLs present and the higher the load of the specific antibody, the greater the risk of cognitive impairment. There is some evidence to suggest that besides the thrombotic process, inflammation-related pathways play a role in the pathogenesis of cognitive impairment in APS. The cornerstone treatments of APS are anti-coagulant and anti-thrombotic medications. These treatments have shown some favorable effects in reversing cognitive impairment, but solid evidence for the efficacy and safety of these treatments in the context of cognitive impairment is still lacking. In this article, we review the current knowledge regarding the epidemiology, pathophysiology, clinical associations, and treatment of cognitive impairment associated with APS and aPL positivity.

## 1. Introduction

Anti-phospholipid syndrome (APS) is an acquired systemic disorder associated with the presence of anti-phospholipid antibodies (aPLs). The classic aPLs include anti-cardiolipin antibodies (aCL), lupus anticoagulant (LA), and the more recently described anti-β2 glycoprotein I antibodies (aβ2GPI) [1]. The primary clinical manifestations of APS are arterial and venous thrombosis, or both, as well as pregnancy morbidity [1]. The presence of at least one aPL antibody without a prior thrombosis or obstetric morbidity determines the presence of an aPL carrier [2]. By contrast, the presence of at least one aPL antibody on two separate occasions, at least 12 weeks apart, accompanied by a history of either a thrombotic event or pregnancy morbidity, is required for the diagnosis of APS [1,3]. APS may occur independently (known as primary APS) or secondary to other autoimmune diseases (known as secondary APS), mainly in the form of systemic lupus erythematosus (SLE) [3].

Neurologic involvement in APS is prevalent and responsible for significant morbidity and mortality [3]. APS may affect the nervous system through several patterns, primarily stroke and transient ischemic attacks (TIA). Besides these more common manifestations, neurologic involvement in APS may manifest as venous sinus thrombosis, cognitive impairment and dementia, psychosis, seizures, movement disorders, headaches, demyelinating syndromes, transverse myelitis, and ischemic optic neuropathy [4]. 

To date, most studies examining cognitive impairment in aPL carriers and in APS have included a small sample size and varied considerably in terms of cognitive impairment detection methods, the particular aspects of cognition evaluated, and the specific antibody type (aCL, LA or aβ2GPI) and the laboratory cutoffs used to define positivity [5]. This complexity in the interpretation of the results is further increased by the following (Table 1): firstly, aPL can be found in the general population with a prevalence of 1–5% [5]. Most of these cases, however, especially if the aPLs are detected at low titers, do not progress to a thrombotic event or cognitive decline, and patients may remain asymptomatic. Thus, aPL carriers represent a highly heterogeneous group of patients, who vary considerably in terms of prognosis and cognitive impairment risk. Secondly, the lack of standardized methods for aPL quantification, which also changes over time, and modifications in the cut-off levels for positivity, lead to further difficulty in comparing the results of different studies. Third, APS can be secondary to autoimmune disease, which may itself affect the central nervous system and cause cognitive impairment. Fourth, aPL antibodies are found in increased frequency in the elderly, among whom cognitive impairment and dementia are common [6]. Thus, the exact frequency and mechanisms of cognitive impairment in APS and their association with aPL activity, as well as the proper approach to diagnosis and treatment, remain unclear [4].

In this review, we summarize the available data regarding the possible associations between cognitive decline associated with aPL and APS. We discuss the epidemiology, pathophysiology, clinical manifestations, and recommendations for management, with an emphasis on the European League Against Rheumatism (EULAR)’s recommendations for the treatment of APS. 

## 2. Epidemiology

### 2.1. Definitions

Dementia is characterized by a decline in cognition from the previous level of function that interferes with daily function and independence. It usually involves one or more cognitive domains (social cognition, complex attention, learning and memory, executive function, language, perceptual-motor). Cognitive impairment is a clinical state between normal cognition and dementia [7]. Several tests have been used for assessing cognitive impairment and dementia, but the Mini-Mental State Examination (MMSE) and the Montreal Cognitive Assessment (MoCA) are most commonly used, with a sensitivity of 75–92% and a specificity of 81–91%, respectively [8].

### 2.2. Carriers of aPL

Cognitive impairment is reported in aPL-positive individuals with a frequency that ranges between 19% to 40% [9,10,11]. A systematic literature review by Bucci et al. reported a prevalence ranging between 5.9% and 31.1% [12]. Among aPL-positive elderly people (>65 years old), an association between high aCL titers (mainly IgG subtype) and cognitive impairment has been reported [13]. In aPL-positive, non-elderly, asymptomatic adults, an increased frequency of cognitive impairment was also documented compared to age and education-matched controls (33% vs. 4%) [9]. An association between aPLs and brain infarction, as well as cognitive and motor decline, was also reported [14]. Among aPL carriers, cognitive dysfunction is much more common than dementia (19–40% vs. 0–6%, respectively), although the latter condition has been reported as well. Supporting the possible role of aPLs as pathogenic factors that can contribute to cognitive decline, two studies demonstrate that aPLs are significantly more prevalent among patients with dementia than in individuals with preserved cognition [15,16].

### 2.3. APS Patients

A correlation between cognitive impairment and high levels of aPL was similarly reported in primary and secondary APS [17,18,19]. However, other studies did not find a significant association between the presence of aPLs and cognitive impairment [20,21]. It is worth noting that all three types of aPL antibody are associated with a risk of cognitive impairment, with the exception of IgM aβ2GPI, which does not confer the same risk. Moreover, the risk of cognitive impairment increases with the number of antibody types detected, and the higher the titer and the persistence over time of the specific aPL, the greater the risk of cognitive impairment [22].

Among primary APS patients, the frequency of cognitive impairment ranges between 42 and 80%, compared to the 7–75% range observed in patients with secondary APS [17,23,24,25,26,27,28,29]. The frequency of dementia and its association with APS have not been rigorously studied in APS patients [5]. The frequency of dementia in primary APS varies widely, ranging from 1.9–2.5% (elderly and non-elderly patients) to 56% (cohort including only elderly APS patients) [30,31,32]. In secondary APS (aPL-positive SLE patients), the frequency of dementia ranges from 21–54% compared to 4–7% in aPL-negative SLE patients, emphasizing once again the possible contributions of aPL to dementia [16,17,18,19,28,29,33,34,35].

Two recent systematic literature reviews reported that the prevalence of cognitive impairment among aPL carriers, primary APS, and secondary APS, when considered together as a group, ranges between 15 and 42% [36,37].

In a very recent study by Sevim et al., the APS ACTION registry, which aimed to describe the baseline characteristics of about 800 patients with aPL positivity, cognitive impairment was reported in 85 (11%) patients [38]. In the group of patients with aPL positivity without APS, 11 (7%) patients had cognitive impairment, while the group of patients with aPL positivity and APS 74 (12%) had cognitive impairment. Furthermore, the prevalence of cognitive impairment among APS patients was higher in patients with thrombotic APS than in patients with obstetric APS, 53 (12%) vs. 3 (4%), respectively. When examining the prevalence of cognitive impairment according to the antibody type and number of positive aPLs, patients with double and triple positivity had higher prevalence than patients with single positivity, 20 (12%) and 33 (12%) vs. 17 (8%), respectively. Moreover, patients with single LAC positivity had a slightly higher prevalence of cognitive impairment compared to patients with non-LAC single positivity, 14 (8%) vs. 3 (6%), respectively.

## 3. Pathophysiology

The pathogenesis of cognitive impairment in APS has not been fully elucidated. Historically, cognitive impairment in APS was mainly attributed to aPL antibody-related microvascular thrombosis resulting from hypercoagulability, with venous and/or arterial thrombosis. The pro-thrombotic effects of aPL through endothelial dysfunction, the activation of platelets, complement, and the coagulation cascade, are well known [3,5]. Antovic et al. proposed that the impairment of the fibrinolysis process may further drive the prothrombotic mechanism in APS [39]. However, more recent studies demonstrated that aPL patients with cognitive decline may have normal brain images on magnetic resonance imaging (MRI), with no evidence of thrombotic events [13]. Thus, cognitive impairment in aPL patients cannot be simply explained by ischemic events.

To explain the possible non-thrombotic contribution of aPL to brain injury, investigators assessed animal models. In a murine model of APS, mice developed neurological and behavioral disorders, including hyperactivity, memory impairment, and aggression, without any evidence of thrombosis [40,41]. Furthermore, the intracerebroventricular injection of aPL antibodies from APS patients directly into mouse brains caused impairment in cognitive performance and the onset of hyperactive behavior [42]. The microscopic examination of brain tissue in mouse models of APS with cognitive impairment revealed a mononuclear inflammatory infiltrate in the choroid plexus and the hippocampus without evidence of ischemic lesions [41,43,44,45]. In vitro studies also showed that aPL antibodies bind to central nervous system neuronal cells [42]. In some studies, the degree of cognitive impairment correlated with aPL titers [41]. The direct binding of aPL antibodies to brain tissue, the relationship between the titer of aPL and the degree of cognitive impairment, and the mononuclear inflammatory infiltrate in brain tissue, all suggest a direct effect of aPL antibodies on brain tissue through an inflammatory mechanism, the nature of which remains to be determined. Moreover, several case reports showed that dementia and other neurologic features of APS may be reversible through the initiation of immunosuppression, emphasizing the possibility of inflammatory pathways acting as driving mechanisms for cognitive impairment [46].

These findings suggest that autoimmune mechanisms may underlie, at least partially, the cognitive impairment observed in patients with aPLs. Thus, cognitive impairment in these patients may actually be a result of the combined effects of hypercoagulability, blood–brain barrier disruption, the activation of pro-inflammatory mechanisms as a result of the direct binding of aPL to brain tissue, and genetic predisposition (Figure 1).

A recent study by Rosa et al. examined the association between brain-derived neurotrophic factor (BDNF), a neuroprotective mediator, and cognitive impairment in primary APS, with lower levels of BDNF associated with cognitive impairment in these patients [47].

## 4. Clinical Manifestations

Cognitive function is a composite of various domains, including perception, motor skills and construction, attention and concentration, memory (including working, declarative, procedural, semantic, and prospective), executive functioning, processing speed, and language/verbal skills. A variety of tools and questionnaires have been developed to assess different aspects of cognition. Each test was developed and tested in different groups of patients and thus has a specific reliability, validity, sensitivity, specificity, and positive predictive value, which together determine the usefulness of each test and the clinical scenario for which it is best suited [48]. Incorporating large batteries of tests (e.g., Wechsler tests) typically provides better cognitive evaluation and good validity.

Table 2 provides a summary of studies evaluating cognitive impairment in aPL carriers and APS patients. It is important to note that these studies employed multiple methods and tests to assess cognitive performance. For instance, Schmidt et al. performed a study on a German-speaking population using validated German language-based tools, while other researchers chose different tests to assess cognition. This variability is understandable but significantly limits comparability across studies.

### 4.1. Cognitive Impairment in aPL Carriers

Multiple cognitive functions are impaired in aPL carriers (Table 2). The main cognitive functions affected are related to executive functioning, working memory, visual and verbal learning, verbal fluency, visuospatial ability, and visuomotor speed and flexibility (Table 2). However, gross attentional processes and fine motor skills appear to be unaffected in aPL carriers (Table 2). No association between any demographic or clinical characteristics and cognitive impairment in aPL asymptomatic carriers has been identified.

### 4.2. Cognitive Impairment in APS

A wide range of cognitive functions are impaired in primary APS (Table 2). The main cognitive functions impaired are related to visual learning, memory, visuomotor and visuospatial speed and flexibility, verbal fluency, and rapid auditory information processing [24]. Tektonidou et al. found that age and livedo reticularis were associated with cognitive impairment in primary APS, while dementia was more common with increasing age and when a greater number of abnormalities were present in electroencephalogram (EEG) and computed tomography (CT) of the brain [24].

SLE is a systemic disease affecting various tissues, including brain tissue, causing neuropsychiatric symptoms and cognitive impairment [49,50]. Kozora et al. compared cognitive function among SLE patients, aPL carriers, and healthy controls using standardized cognitive assessment tests, and brain (with functional) MRI. They showed that aPL carriers and SLE patients had abnormal brain activity, mainly affecting the frontal cortex, even when no overt clinical symptoms of cognitive decline were present [51]. This finding suggests that both conditions, aPL carrier and SLE, probably contribute to cognitive impairment. Similarly, Ilgen et al. showed that the coexistence of aPL antibodies with SLE (secondary APS SLE-associated) increases the risk for neurologic involvement compared to patients diagnosed with SLE without APS [52].

The main cognitive functions impaired in patients with secondary APS (mainly SLE-associated) are related to executive functions, complex attention, verbal memory and verbal fluency, as well as visuospatial domain [24]. Tektonidou et al. reported cognitive decline in both primary and secondary APS (SLE-associated) compared to healthy controls, with no differences between primary and secondary APS patients [24]. The presence of aPL antibodies, hypertension, and a history of stroke were previously described as factors associated with cognitive decline in SLE-associated secondary APS [29]. The presence of aPL antibodies was also demonstrated as an independent predictor for cognitive decline in SLE patients in other studies [18,19]. Gomez-Puerta et al. reported a total of 30 (25 cases from published research and 5 cases from their own cohort) patients with dementia and APS (14 had primary APS, 9 had APS secondary to SLE, and the rest had “lupus-like” disease). The authors concluded that even though dementia is not common in patients with APS, it causes significant disability [49]. A prospective study evaluated the neuropsychiatric manifestations of APS, including cognitive decline, in 1000 SLE patients with no evidence of previous strokes. In this study, as in the previous studies, the presence of aPL antibodies was found to be a predictor for the development of cognitive decline. However, this finding lost statistical significance when thromboembolic events were excluded, suggesting that, at least in this latter study, the association of cognitive decline with aPL antibodies was mostly related to thrombotic pathways rather than inflammation incited by aPL antibodies [20]. Erkan et al. analyzed the functional outcome in 39 patients with primary APS over 10 years. About 20% of these patients were functionally impaired, mainly due to cognitive dysfunction [53].

## 5. Treatment

Long-term anticoagulation with oral warfarin is a cornerstone treatment in APS patients presenting with thrombotic lesions and neurologic manifestations, including cognitive impairment [54]. Hughes reported that even features such as headache and memory loss improved with appropriate warfarin dosage [55]. The role of direct oral anticoagulant therapy for the secondary prevention of stroke in APS is still under evaluation. The ongoing Rivaroxaban in Stroke Patients with APS (RISAPS) trial aims to compare the efficacy of high-intensity rivaroxaban 15 mg twice daily versus warfarin in the prevention of secondary stroke in APS patients [56]. Nevertheless, because of our limited understanding of the pathogenesis of cognitive impairment contributed by non-thrombotic pathways, the efficacy of immunosuppressive treatment has not been sufficiently studied [57]. Thus, no therapeutic guidelines are available for the treatment of cognitive impairment in aPL carriers and APS patients presenting with cognitive impairment without thrombotic lesions seen by brain imaging [58]. The “RITAPS” study was the first formal attempt to investigate the role of immunosuppression in the management of cognitive impairment [57]. This study was a randomized clinical trial designed to evaluate the efficacy of rituximab on non-criteria aPL manifestations, including cognitive impairment, in aPL carriers as well as APS patients. Patients received two doses of rituximab (1000 mg two weeks apart). The efficacy was evaluated immediately after the administration of rituximab and at 24 through 52 weeks. Cognition was evaluated using 12 tests from a standardized neuropsychological test battery endorsed by the ACR. In this study, only six patients had cognitive impairment at baseline, of whom three had a complete response, one had a partial response, and one had no response; one patient terminated the study early. The main improvements were observed in the attention, visuomotor speed, and flexibility domains. While potentially interesting, the limitations of the RITAPS study were evident and included its small sample size and relatively short follow-up period. However, the study was well designed and prospective, and it demonstrated clear benefits in the cognitive domain in association with treatment.

The European League Against Rheumatism (EULAR)’s recommendations for the management of anti-phospholipid syndrome in adults recommend the use of low-dose aspirin for primary prevention (e.g., for the purpose of preventing the first thrombotic event) in aPL carriers with high-risk aPL profiles (defined as any of the following: multiple aPL positivity, lupus anticoagulant or persistently high aPL titers), without addressing cognitive impairment specifically [58]. Interestingly, McLaurin et al. found that consistent aspirin use was associated with improved cognitive function in older patients with SLE [33]. Hence, there is not sufficient evidence to support the use of aspirin as a primary prevention therapy for preventing cognitive impairment in aPL carriers beyond its use to prevent thrombotic events, as per this particular set of recommendations.

To date, no clinical studies have directly examined the effect of hydroxychloroquine (HCQ) on cognitive impairment in APS. HCQ is an anchor therapy in SLE, and many studies have shown favorable effects on damage accrual and survival in SLE patients [59,60,61,62]. However, these studies have not directly examined the accrual of NPSLE damage standing alone, but rather as a component of more complex composite damage scores. Groot et al. showed that HCQ was associated with an absence of damage according to the Systemic Lupus International Collaborating Clinics/American College of Rheumatology Damage Index (SDI), which includes neuropsychiatric disease [60]. Similar results were reported by Fessler et al. in the LUMINA study [61].

Ceccarelli et al., in their cohort study, evaluated changes in SLE-related cognitive impairment over 10 years. They found that cognitive impairment improved in the majority of patients, but the use of HCQ, or immunosuppressants, was not associated with a change in cognitive impairment over time [63]. Another cohort study assessed the risk of dementia in connective tissue disease patients and found no significant difference in the risk of dementia among long-term HCQ users compared to non-users [64]. Unrelated to lupus or APL, but perhaps relevant to the question of whether HCQ has neuroprotective properties, a randomized controlled trial on Alzheimer disease, which examined the effect of HCQ on progression to dementia, found no significant effect [65]. While interesting, the relevance and applicability of these studies to the question of whether HCQ is protective against cognitive dysfunction in aPL patients is limited and/or circumstantial at this point, since these studies were not designed to address this question directly.

Some case reports have demonstrated the successful treatment of aPL-associated cognitive dysfunction or neurological manifestations with mycophenolate mofetil and immunoglobulins, but properly designed prospective studies are needed to confirm their efficacy [66].

## 6. Conclusions

In summary, while cognitive impairment in APS is less studied than other manifestations of the syndrome, this type of neurologic involvement seems to be relatively common among APS patients, as well as aPL antibody carriers. The exact prevalence is unclear, mainly due to inconsistency in the tools used to identify impairment in these studies. The higher the number of the aPLs and the higher the load of the specific antibody, the greater the risk of cognitive impairment. Some evidence suggests that besides the thrombotic process, inflammatory injury plays a role in the pathogenesis of cognitive impairment in APS. The cornerstone in the treatment of APS is anti-coagulant and anti-thrombotic modalities. These treatments showed some favorable effects in reversing cognitive impairment (when there is another accepted indication for treatment), but solid evidence for the efficacy and safety of these treatments in the context of cognitive impairment is, unfortunately, still lacking.

## 7. Research Agenda

Several questions remain unanswered in the context of cognitive impairment in aPL carries and patients with APS. First, can low aspirin prevent cognitive impairment in aPL carriers? This question may be best answered by comparing patients treated with aspirin with untreated patients by following them prospectively for the occurrence of cognitive decline. Second, are anti-coagulant and anti-thrombotic therapies effective in the management of cognitive impairment in patients with APS? Third, is there a place for immunosuppressive therapy in treating cognitive impairment in patients with APS? Fourth, is there any effect of the cumulative duration of aPL positivity on the severity of cognition defects in APS and aPL-carriers? These are only some, but very challenging, questions that need to be addressed in the coming years.

## Figures and Tables

**Figure 1 brainsci-12-00222-f001:**
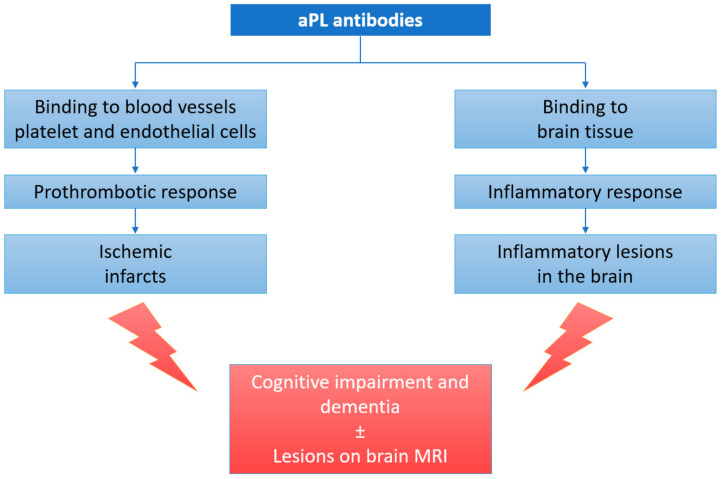
Pathophysiology of cognitive impairment in aPL carriers and APS patients. aPL—anti-phospholipid antibodies; MRI—magnetic resonance imaging.

**Table 1 brainsci-12-00222-t001:** Factors limiting the comparability across studies of cognitive impairment in aPL carriers.

aPL can be found in the general population, but most cases are not associated with cognitive decline.
aPL quantification is performed using various methods.
Cut-off levels for positivity for aPL have changed over time.
APS can be secondary to autoimmune disease, which may itself affect cognition.
The prevalence of aPL antibodies increases in the elderly population, in which cognitive impairment and dementia are also common.

aPL—anti-phospholipid antibodies; APS—anti-phospholipid syndrome (APS).

**Table 2 brainsci-12-00222-t002:** Cognitive impairment in aPL carriers and APS patients.

Study	Study Population	Control Group	Cognitive Tests	Main Results	Cognitive Impairment Frequency
**aPL Carriers**					
Schmidt et al., 1995 [13]	Elderly subjects (*n* = 53)	Age-matched non- aPL carriers	MWT-B, Janke and Debus, LGT-3, WCST, Alters Konzentrations of Gatterer, Purdue Pegboard	Impaired memory and visuopractical abilities. No brain abnormalities or differences in brain MRI.	Not reported
Jacobson et al., 1999 [9]	Asymptomatic, aPL-carriers, non-elderly adults (*n* = 27)	Age- and education-matched non-aPL carriers	Wechsler, CVLT, Benton line orientation, COWAT, finger oscillation, grooved pegboard, RCFT, trail making, WCST, Beck, state-trait anxiety inventory	Impaired executive functioning, verbal learning, memory, and visuospatial abilities. Attentional processes and fine motor skills appeared unaffected.	33% in aPL carries vs. 4% in controls
Erkan et al., 2010 [10]	High titers of aPL antibodies (*n* = 85)	Moderate titers of aPL antibodies (*n* = 58)	Not specified	Increased prevalence of cognitive impairment in the higher-titer group in a linear pattern	12% in high titers vs. 3% in moderate titers group
Kozora et al., 2014 [11]	Non SLE aPL-carriers (*n* = 20)	SLE patients with negative aPL	FSIQ, Wechsler digit symbol and block design, trail making, Stroop color and word, CVLT, Rey-O Immediate, Rey-O Recall, LNST, COWAT, PASAT, Dig Vig, category test, finger tapping test	High frequency of cognitive impairment in both groups with no significant difference between the groups	40% in non-SLE aPL carriers vs. 60% in the SLE non aPL carriers
**Primary APS**					
Tektonidou et al., 2006 [24]	Primary APS (*n*= 39) and secondary APS (SLE related) (*n* = 21)	Healthy age-, sex-, and education-matched controls	Wechsler digit span, symbol and block design, Rey AVLT, RCFT, SCWT, TMT, COWAT.	Impairment of visual learning, memory, visuomotor and visuospatial speed and flexibility, verbal fluency, and rapid auditory information processing impaired. No difference between primary APS and secondary APS Predictors for cognitive impairment: Livedo reticularis and presence of white ma ter lesions on MRI	42% in APS patients vs. 18% in the controls
Coin et al., 2015 [23]	Primary APS (*n* = 15), Secondary APS with SLE (*n* = 12) and SLE without aPLs (*n* = 27)	Healthy, age- and education-matched controls	TAVEC, RCFT, Stroop color and word test, verbal phonemic fluency and semantic fluency (Spanish version), Ruff 2&7 selective attention test.	Impaired executive functions and memory (verbal and visual)	80% in primary APS, 75% in secondary APS with SLE, 48% in SLE without aPLs, and 16% in the controls
**Secondary APS**					
Maeshima et al., 1992 [25]	Secondary APS with SLE (*n* = 21)	Healthy controls	MMSE, “Kana” pick-up test, Miyake’s paired associated memory scale, word recall, digit span, Watamori method, line bisection test, line cancellation task, recognition of intricate pictures and perspective cube copying test.	Higher cortical impairment in study group	76% vs. missing data
Afeltra et al., 2003 [17]	Secondary APS with SLE (*n* = 61)	Healthy controls	Not specified	High titers of aPL were associated with cognitive impairment No details on cognitive impairment patterns	58%
Mikdashi et al., 2004 [18]	Secondary APS with SLE (*n* = 130)		MMSE with other tests not explicitly specified	No details on cognitive impairment patterns	27% in study group
McLaurin et al., 2005 [33]	Secondary APS with SLE (*n* = 123)		Mild impairment battery from the Automated Neuropsychological Assessment Metrics (ANAM).	No details on cognitive impairment patterns	37.5% in study group
Tomietto et al., 2007 [28]	Secondary APS with SLE (*n* = 52)	Rheumatoid arthritis	Raven’s progressive matrices, comprehension, similarities, block design, and digit symbol of Wechsler, Wechsler memory scale, Rey auditory-verbal learning, trail-making, Corsi block, number cancellation, reverse numerical sequence (MMSE), Stroop word and color test, semantic and phonemic verbal fluency, denomination of Aachener Aphasie and token test.	Executive functions and complex attention were more frequently impaired in APS patients.	68.6% in study group vs. 41.2% in controls
Murray et al., 2012 [29]	Secondary APS with SLE (*n* = 694)		HVLT-R, COWAT	Verbal memory and verbal fluency	15% in entire cohort
Conti et al., 2012 [19]	Secondary APS with SLE (*n* = 58)		Standardized testing from ACR and the CSI standardized in an Italian population	Visuospatial domain mainly impaired	Missing data

aPL—anti-phospholipid antibodies; APS—anti-phospholipid syndrome; MWT-B—mehrfachwahlwortschatztest; LGT-3—Bäumler’s Lern-und Gedächtnistest; WCST—Wisconsin card sorting test; MRI—magnetic resonance imaging; SLE—systemic lupus erythematosus; CVLT—California verbal learning test; COWAT—controlled oral word association test; RCFT—Rey complex figure test; FSIQ—full-scale intelligence quotient; Rey–O immediate, Rey–O recall—immediate and 30-min delayed recall of the Rey–Osterrieth complex figure test; LNST—Letter-–number sequencing test; PASAT—Paced auditory serial addition test; Dig Vig—digit vigilance test; Rey AVLT—Rey auditory verbal learning test; SCWT—Stroop color–word interference test; TMT—trail-making test; TAVEC—Spanish version of the California learning verbal test; MMSE—mini-mental-state examination; HVLT-R—Hopkins verbal learning test—revised; ACR—American college of rheumatology; CSI—cognitive symptoms inventory.

## Data Availability

Not applicable.

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
