# Peer review of "Cognitive Impairment in Anti-Phospholipid Syndrome and Anti-Phospholipid Antibody Carriers"

_brainsci, 2022, doi:10.3390/brainsci12020222_

Round 1
Reviewer 1 Report
This is a narrative review. The main aim of the authors was to summarize the prevalence and pathogenesis of cognitive impairment in anti-phospholipid syndrome (APS) and anti-phospholipid antibodies (aPL) carriers, providing a rationale for the treatment of this neglect manifestation of the disease. The article is simple and clear, global readability is good. However, the bibliography review is not fully updated. I think this article could have a place in APS literature if some modifications are provided. I have some comments.
Major comments
“Epidemiology”: at least two systematic literature reviews (SLRs) were performed in the last 3 years to investigate this topic. These works should be cited in the work, even if the primary focus of these SLRs was partially different from the one of this narrative review (DOI: 10.3233/JAD-181294; DOI: 10.1093/rheumatology/keab452). A careful comparison of the results of these studies should be provided by the authors.
“Pathophysiology”: a recent cross-sectional study evaluated the association between serum levels of brain-derived neurotrophic factor and cognitive dysfunction in APS, suggesting a role as a possible biomarker for this serum indicator. The authors should update their review including this research article (DOI: 10.1093/rheumatology/keaa252).
Minor comments
Abstract: “The cornerstone in the treatment of APS is anti-coagulants and anti-thrombotic medication”. There is a typo. Please, correct.
“Introduction”: “In contrast, the presence of at least one aPL antibody, accompanied by a history of either a thrombotic event or pregnancy morbidity, fulfils the diagnosis of APS”. I would stress that the aPL positivity must be confirmed in two determinations.
“Introduction”: “Most of the studies to date examining cognitive impairment in aPL carriers and in APS included a small sample size, and varied considerably in terms of cognitive impairment detection methods”. I would suggest including here a comparison with the cognate condition Systemic Lupus Erythematosus (SLE), in which, similarly, cognitive impairment is extensively studied but with a great variety of tools and questionnaires to detect such involvement, limiting the comparability across studied.
“Introduction”: I suggest including a table or a vignette to summarize the main limitations in the generalizability of cognitive impairment assessment in APS and aPL carriers.
“Pathophysiology”: “If this is indeed the case, cognitive impairment in these patients may actually a result of a combination of hypercoagulability, blood-brain barrier disruption, activation of pro-inflammatory mechanisms a result of direct binding of aPL to brain tissue, and genetic predisposition”. There is a typo. Please, correct.
“Clinical manifestations”: I would include a table reporting the tools and questionnaires used to detect cognitive impairment in APS and aPL-carriers. Which was the most frequently adopted tool? Is there a rationale to support the adoption of a specific tool in cognitive impairment assessment in the context of APS?
“Clinical manifestations”: the authors do not discuss on whether disease duration (e.g. timespan from first detection of aPL to cognitive dysfunction diagnosis”) have an influence on cognition defects in APS and aPL-carriers. Please, discuss briefly on it if applicable.
“Treatment”: could the authors retrieve some data supporting an effect of hydroxychloroquine in addressing cognitive impairment in APS? Could they find some examples in other diseases, like SLE?
Reviewer 2 Report
This is a nice and well written narrative review addressing the issue of cognitive dysfunction related to the antiphospholipid syndrome. The topic is of interest given the uncertainty about epidemiology, pathophysiology and treatment.
I have no major issue to raise, but only some suggestions aimed to improve the quality of the review:
- TITLE : As the review address not only the relationship between cognitive impairment and APS but consider aPL carriers too, I suggest to modify the title accordingly
- EPIDEMIOLOGY : two recent SLR about a similar topic have been published, and could be included as references and discussed as well:
- Donnellan C et al. Rheumatology 2021 doi: 10.1093/rheumatology/keab452.
- Bucci T et al. J Alzheimers Dis. 2019;69(2):561-576. doi: 10.3233/JAD-181294.
- TREATMENT : The Authors opportunely quote the RITAPS pilot study. I suggest to mention in the text also the total number of patients included in this phase II pilot study (19 pts). As correctly reported, 6 patients had cognitive impairment at baseline, but the Authors report - in the review - the outcome of only 5 of them, missing the 6th patient (marked in the original paper as ET: early termination). This information - for completeness - should be noticed in the text.
Reviewer 3 Report
The authors prepared a narrative review about cognitive impairment in APS/aPL-positive patients. Review articles (some of which were systematic) tackling this subject were published this year, therefore the scientific value of the current manuscript is limited. I urge the authors to follow systematic review guidelines as well as attempt to answer more specific scientific questions if they choose to modify this manuscript.
This statement/conclusion is only partially supported by the literature evidence included in this manuscript:
" If this is indeed the case, cognitive impairment in these patients may actually a result of a combination of hypercoagulability, blood-brain barrier disruption, activation of pro-inflammatory mechanisms a result of direct binding of aPL to brain tissue, and genetic predisposition (Figure 1)."
Round 2
Reviewer 1 Report
All my comments were answered by the authors.
Author Response
Thank you very much.
Reviewer 3 Report
Thank you for modifying the manuscript
Author Response
Thank you very much.